# Effectiveness of a Combined Intervention on Psychological and Physical Capacities of Frail Older Adults: A Cluster Randomized Controlled Trial

**DOI:** 10.3390/ijerph16173125

**Published:** 2019-08-28

**Authors:** João Apóstolo, Maria dos Anjos Dixe, Elzbieta Bobrowicz-Campos, Timóteo Areosa, Rita Santos-Rocha, Mónica Braúna, Jaime Ribeiro, Isabel Marques, Joana Freitas, Maria de Lurdes Almeida, Filipa Couto

**Affiliations:** 1Nursing, Nursing School of Coimbra (ESEnfC), The Health Sciences Research Unit, a Collaborator of The PORTUGAL Centre for Evidence-Based Practice: A Joanna Briggs Institute (JBI) Centre of Excellence (PCEBP), 3046-851 Coimbra, Portugal; 2Centre for Innovative Care and Health Technology (ciTechCare), Polytechnic Institute of Leiria (IPLeiria), 2411-901 Leiria, Portugal; 3Nursing, Nursing School of Coimbra, The Health Sciences Research Unit, 3046-851 Coimbra, Portugal; 4Sports Sciences School of Rio Maior (ESDRM), Polytechnic Institute of Santarém (IPSantarém), 2040-413 Rio Maior, Portugal; 5Interdisciplinary Centre for the Study of Human Performance-Faculty of Human Kinetics-University of Lisbon, 1499-002 Cruz-Quebrada, Portugal

**Keywords:** older adults, frailty, combined intervention, cognitive stimulation, Animal-Assisted activities, physical exercise, depressive symptomatology, gait analysis

## Abstract

Background: Older adults experience physical and psychological declines affecting independency. Adapted and structured combined interventions composed of cognitive stimulation and physical exercise contribute to comorbidities’ reduction. *Methods:* Multicenter single-blinded two-arm cluster randomized controlled trial conducted to assess effectiveness of a combined intervention (CI), composed of a cognitive stimulation program (CSP) and a physical exercise program (PEP), on psychological and physical capacities of frail older adults as to on their activities of daily living. Were recruited 50 subjects from two elderly end-user organizations. Of these, 44 (65.9% females, mean age of 80.5 ± 8.47 years) were considered eligible, being randomly allocated in experimental (EG) or control group (CG). Data collected at baseline and post-intervention. EG received CI three times a week during 12 weeks. CG received standard care. Non-parametric measures were considered. *Results:* At baseline, groups were equivalent for study outcomes. The comparison of pre- and post-intervention data revealed that subjects receiving CI reduced depressive symptomatology and risk of fall based on gait and balance, and improved gait speed. Simultaneously, in the CG a significant decline on activities of daily living was observed. Significant results were found among biomechanical parameters of gait (BPG). EG’ effect size revealed to be small (0.2 ≤ *r* < 0.5). CG’ effect size was also small; but for activities of daily living there was an evident decrease. *Conclusion:* The CI is effective on managing older adults’ psychological and physical capacities.

## 1. Introduction

According to the World Health Organization (WHO), by 2030, it is expected that more than 75 million people will experience dementia. The adverse health-related outcomes associated with dementia increase older adults vulnerability and are now a public health priority, once they increase long-term care costs and negatively impact governments, communities, families, and individuals’ autonomy [1]. There is growing evidence suggesting that frailty may increase the risk of future cognitive decline, thus representing a novel modifiable target in early dementia [2]. The promotion of active aging policies guided by independent living, quality of life improvement, and health care costs reduction [3] may significantly reduce older adults’ frailty and functional dependency, and contribute to the maintenance of physical and cognitive capacities for as long as possible [4].

Different models and concepts are presented in the literature for frailty. Some of them focus exclusively on the physical domain, as proposed by [5], defining frailty in terms of phenotypic markers, such as global weakness with low muscle strength, overall slowness (especially in gait), decreased balance and mobility, fatigability or exhaustion, low physical activity, and involuntary weight lost. However, according to some authors, a definition just oriented to physical changes is not enough to define frailty [6]. Another accepted definition considers frailty as an age-related state of decreased physiological reserves characterized by a weakened response to stressors, associated with numerous pathophysiological modifications in different body systems. This vulnerability in different physiological systems induces cognitive and physical impairments, increasing the risk of development and onset of geriatric syndromes [7]. Other authors went further and added, beyond the cognitive and physical components, the mood-related and social components. Thus, frailty is defined as a dynamic state that affects a person who experiences losses in their psychological (including cognition and mood-related aspects), physical, and/or social domains. This state may be caused by numerous variables and increases the risk of adverse health-related outcomes [7].

The lack of consensus on the frailty definition has a negative impact not only on screening and assessment methods, but also on interventions and clinical approaches to frailty, which enable the use of uniform tools to its diagnosis and management. Yet, it seems quite consensual that, due to frailty’ complexity, a multidimensional approach to its management is required. Another common point among models and concepts is related with the identification of a possible potential to reverse frailty [8]. As a mediator of diseases, frailty requires the adoption of strategies and actions that prevent adverse health-related outcomes, and allow an early identification of individuals at risk for early implementation of prevention and intervention techniques [9].

Frailty assessment methods still do not demonstrate valid, reliable, and diagnostic accuracy mainly due to the above mentioned lack of consensus and is the main reason why a specific frailty assessment tool was not considered or used in this study. In fact, an overview of reviews concluded that there are no universally appropriate specific screening tools to identify frailty and the choice of assessment tools should be based on the context and purpose for which it is needed [10]. In this way, for the study purpose, and considering the study context, were established isolated outcomes related to physical and psychological domains, by having as theoretical basis the Gobbens’ Conceptual Model for Frailty [7]. This trial considered then a multidimensional approach and has its focus on physical frailty components (balance, mobility, and physical exercise), psychological frailty components (cognitive function and depressive symptomatology), and socio-demographic data.

### Intervention Rational

Physical and cognitive interventions are related with an improvement of older adults’ physical and psychological capacities that due to the ageing process are frequently decreased [11]. When singly implemented, these non-pharmacological interventions improve health-related outcomes, namely those that are related with functionality and autonomy. Physical exercise is also associated with higher cerebral blood flow [11] that contributes to the optimization of cognitive performance.

A systematic review of the effectiveness of interventions to prevent frailty progression in older adults has identified as effective physical and cognitive programs delivered in a group [12]. Positive effects were also found on multicomponent treatments such as when physical, cognitive, and nutritional programs are combined [12]; however, there are only a few studies that have examined the impact of an adapted and structured combined intervention composed of cognitive and physical interventions for frailty.

A sedentary lifestyle among older adults leads to an increased decline of physical capacities [13]. Physical exercise interventions benefit older adults’ functional capacity [14]. In the specific context of frailty, physical exercise programs oriented for resistance training at a moderate-to-high intensity level, are associated to improvements in the muscle mass, bone density, and cardiometabolic, which may reflect earnings on functional health status, mental health, and cognitive performance [4]. Evidence has shown that physical exercise has value as a strategy to reduce the risk of cognitive decline in older adults once it acts on modifiable risk factors, which is supported by the possibility of cognitive decline being delayed [15].

Cognitive stimulation is described as an intervention that aims to promote involvement in activities focused on enhancement of cognitive and social functioning [16]. It should include structured programs that provide cognition training through teaching skills and strategies for improving functioning in one or more cognitive domains. These programs should also be based on activities of daily living, allowing for the transfer of trained abilities to the context of everyday life [16]. Evidence has revealed that cognitive training-based interventions, provided to community-dwelling pre-frail and frail older adults have positive impact on cognitive frailty [12]. Gait analysis activities may be included as a component of cognitive interventions for older adults with cognitive decline. They had shown positive influence on cognition, as they stimulate memory [17] and improve social interaction [18].

Animal-assisted activities in a health context refer to a type of intervention that intentionally includes activities with animals (especially dogs) in order to promote benefits in health and well-being of individuals and groups. Regarding institutionalized older adults, it seems that this type of intervention promotes improvements in their perceived quality of life [19,20], emotional state [18], and stimulates positive emotions [21].

Within all this complexity, the solution to reduce and to manage the prevalence of frailty goes through implementing multidomain and preventive nonpharmacological interventions that are suggested to be more effective in frailty context [12,22,23]. These interventions include physical activity or exercise, cognitive stimulation and training, promotion of healthy dietary habits, and promotion of resilience [22]. Physical exercise interventions, when combined with cognitive stimulation may have an impact on psychological and physical capacities of frail older adults. Based on the above arguments the research team decided to evaluate the effectiveness of a combined intervention on health-related outcomes in frail older adults. For this purpose, impact of combined intervention (CI) provided to the experimental group (EG) was compared with an impact of standard care delivered to the control group (CG). It was hypothesized that, in comparison to the participants from the control arm, the participants from the experimental arm at post-intervention would have better outcomes in terms of cognitive function, depressive symptomatology, gait speed, biomechanical parameters of gait, and risk of fall based on gait and balance. Besides, it was also hypothesized that CI would improve activities of daily living of frail older adults.

## 2. Materials and Methods

### 2.1. Aims, Design and Setting

A single-blinded, two-arm cluster randomized controlled trial (RCT) with a multicenter approach was conducted in the Center region of Portugal in two elderly end-user organizations (two clusters each), during 2018. This RCT was registered with ClinicalTrials.gov (identifier: NCT03390478).

The study main aims were: To assess the effectiveness of a CI composed of cognitive stimulation program (CSP) and a physical exercise program (PEP) on psychological and physical capacities of older adults as to on their activities of daily living.

### 2.2. Recruitment and Sampling

The present multicenter study targets Portuguese frail older adults from two elderly end-user organizations day-centers and nursing homes) from the project consortium (Caritas Diocesana de Coimbra and Santa Casa da Misericórdia de Alcobaça). The principal manager of each elderly end-user organization will provide to the research team possible day centers or nursing homes to participate in the study, from which potential participants were screened. To be eligible for the study, elderly end-user organizations should not offer as standard care the adapted, structured, and validated cognitive stimulation programs or physical exercise programs. This criteria was stablished once most of the delivered programs were not structured as a combined intervention, and were not submitted to a validation process within the specific population, lacking evidence of its effectiveness.

Older adults were eligible if they met the following inclusion criteria:(i)aged 65 years or above;(ii)the ability to consent their participation in the study in an informed manner;(iii)the presence of medical clinical conditions that allow them participation on CI;(iv)without severe cognitive impairment as per six-item Cognitive Impairment Test (6-CIT) total score equal or higher than 21 original version of [24], adapted and validated to Portuguese by [25];(v)without severe depressive symptomatology, using the Portuguese version of the Geriatric Depression Scale-five (GDS-5) [26,27], a short version of the GDS-30 developed originally by [28,29]. Total score lower than 2 points [27];(vi)without severe risk of fall tracked by Tinetti Index [30] adapted to Portuguese [31] total score higher than 19 points [31].

Participants were excluded if they presented at least one of the following criteria: (i) lack of a stable clinical condition (such as acute pain or recent surgery) and (ii) lack of desire to participate in the study. An umbrella review [10] highlighted that there are no universally appropriate specific screening tools to identify frailty. Based on the previous arguments a large number of instruments are needed to screen frailty and this procedure was not possible to assess eligibility criteria. It will enlarge the instrument’s battery and overload the participants.

Fifty potential participants, identified by the principal manager of each elderly end-user organization, were screened for eligibility. Of these, six individuals did not meet the inclusion criteria due to high cognitive impairment, lack of a stable clinical condition, and age. Participants from the EG and CG received similar instructions and information about their involvement in the study. During the intervention program, five participants were lost because they passed away (n = one) or worsened their clinical condition (*n* = four). Figure 1 shows the eligibility, random allocation, and follow-up of the study participants.

### 2.3. Randomization and Assessment

Due to the multicentric approach of the study, and in order to prevent contamination between groups, the subject allocation was based on a cluster randomization method. From the four clusters two were randomized to EG and two to CG. To decrease a potential selection bias, an independent researcher, who was not involved in the recruitment process, randomized those clusters using a computer-generated software (https://www.random.org/).

The principal researcher was not blinded to group allocation and informed each elderly end-user organization about in which arm the study participants were allocated. However, to avoid assessment bias, the outcome assessors and data analysts were blinded to the group allocation.

To decrease contamination, CG participants were asked not to carry out activities similar to intervention program. CG participants were integrated on a waiting list to then receive CI.

The trial period counting from the time of recruiting until the final assessment was 14 weeks (Figure 2). At each elderly end-user organization data was collected at baseline and at post-intervention. Two recruiters recruited patients and obtained their consent, assigned the identification codes and collected sociodemographic data. That was collected using an instrument built for this study purpose by the research team. Outcome assessment was done by trained and expert outcome assessors that participated in training meetings. To assess outcomes only adapted and validated Portuguese instruments were used.

### 2.4. Intervention

A CI composed of a CSP and a PEP was implemented over 12 weeks on the EG. Each week, participants received two sessions of physical exercise and one session of cognitive stimulation. Participants from CG received the end-user organization standard care. This included standard activities and usual care provided by the institution and did not include adapted and structured cognitive stimulation programs or physical exercise programs. Before implementation, a meeting was conducted to present all CI components to the clinical managers of the end-user organizations to ensure that they engage in the study. The Figure 3 shows CI components.

#### 2.4.1. Cognitive Stimulation Component

The cognitive stimulation component comprised 12 group sessions, with one session per week: 10 of the cognitive stimulation program (60 min each session) and two of animal-assisted activities (30 min each session). These sessions stimulate different cognitive domains; promote socialization and engagement of older adults, and foster the sense of belonging to a group.

The animal-assisted activities sessions are defined as informal interactions conducted for motivational, educational, and recreational purposes and are associated to have positive impact on the physical and psychological domains [32,33]. Sessions were implemented by a qualified and certified dog handler in accordance with the guidelines proposed by [34]. In this specific case, the dog handler was a nurse specialized in mental health, certified for implement this type of interventions. The same qualified nurse and dog provided this intervention in the two elderly end-user organizations to ensure an equal intervention in both places. Basic hygienic measures after dealing with the dog were ensured. During sessions, the nurse explored different cognitive domains such as memory and language. Two group sessions of animal-assisted activities were so, introduced as cognitive stimulation.

The other 10 cognitive stimulation sessions comprehended the CSP making a difference, originally developed by [35], adapted for the Portuguese context by [36]. The program was adjusted for the current study purpose and was delivered by trained health and social professionals (nurses and occupational therapists).

#### 2.4.2. Physical Exercise Component

The PEP included 24 group sessions (two sessions per week), with a duration of 30 min each one. The PEP aiming at delaying the functional decline in institutionalized frail older populations was designed by exercise specialists to be implemented by healthcare professionals [37]. The exercise program includes several components of posture stability, balance, and strength training that can be adjusted to the context and to the characteristics of the target population. It includes a portfolio of exercises in different support materials. In the current study, qualified exercise professionals delivered the PEP.

### 2.5. Instruments

All assessors were experts in the respective outcome assessment. However, to ensure uniformity in the application of the assessment instruments, a standardized application guide was built and one training meeting of the assessors occurred. All the instruments were applied at baseline and post-intervention, exception made for sociodemographic data that were collected only at baseline. All instruments were used with permission of the instrument ownership companies or authors.

#### 2.5.1. Sociodemographic Data

The research team developed a questionnaire to collect the sociodemographic data such as sex; age, civil state, level of education, and registration time in the end-user organization (in months).

#### 2.5.2. Primary Outcomes

*Cognitive function*—Montreal Cognitive Assessment (MoCA) was used to assess participant’s cognitive function, original version [38], validated to Portuguese by [39]. The MoCA assesses functioning in visuospatial and executive domains, memory, attention, language, and orientation. Higher scores are associated with less cognitive decline.

*Depressive symptomatology*—To assess participant’s depressive symptomatology, the Geriatric Depression Scale with 10 items (GDS-10) was used. This instrument was developed based on the original version of [28,29] and validated to Portuguese [27]. Total scores equal or higher than two points indicates that individuals could be depressed [27].

*Gait Speed*—To assess participant’s gait speed, the gait speed (GS) test with a stopwatch was used. The adopted protocol was based on the directives of [40]. The assessed walking distance was five meters. A 2.5 m distance was established as preparatory and recovery distance. Participants were asked to perform the test with their usual pace. Gait speed less than 0.8 m/s was considered a predictor of poor health status, and sub-clinical neurological and muscular impairment, as suggested by [41].

*Risk of fall based on gait and balance*—The Tinetti Index (TI) was used to assess participant’s risk of fall based on gait and balance. This instrument assesses static and dynamic balance and was originally developed by [30], Portuguese version of [31]. The total score varies between 0–28 points, with low scores being associated to a diminished balance’ capacity and increased risk of fall.

*Biomechanical parameters of gait*—This outcome included the total maximum force and peak pressure of the plantar area. To assess changes from baseline in participant’s performance the Novel EMED-X pressure platform and software were used, in accordance with [42,43].

#### 2.5.3. Secondary Outcome

*Activities of daily living*—To assess participant’s activities of daily living the Barthel Index (BI) was used, original version of [44]. The version of BI used in this study was translated and adapted to Portuguese [45]. The BI considers dimensions such as bathing, grooming, dressing, feeding, toilet use, urinary and fecal incontinence, transferring, walking 50 m, and stair use, with score ranging from 0 (dependence) to 100 (independence). For the study purpose, proportions were considered, so punctuation is presented between 0 (dependence)–1 (independence).

### 2.6. Statistical Analysis

For sample size calculation, primary outcomes were considered and the software G*Power 3.1.9.2. was used, based on literature [46,47] (44 old adults were randomized: 23 were used for the IG and 21 for the CG)

Data were manually transcribed from paper records to a database. To ensure data quality and accuracy, data entry was checked by two researchers blinded to the process. Data normality distribution was analyzed through Shapiro-Wilk test. Baseline performance and differences in sociodemographic data were analyzed using Pearson’s Chi-square test and Mann-Whitney test for nonparametric samples. Non-parametric tests such Mann-Whitney test (U) and Wilcoxon test (Z) were performed to compare outcomes between and within groups. Statistical significance was set at *p* < 0.05. Effect size was calculated using the formulas *r* = Z/√N (Wilcoxon) and Π^2^ = Z^2^/N − 1 (Mann Whitney) [48]. Results were interpreted according [49], who considers the existence of small size if differences are between 0.2 ≤ *r* < 0.5; evidence of average effect size if differences are between 0.5 ≤ and < 0.8 and very high effect size if differences are ≥0.8.

Participants’ results were analyzed in the groups where they were initially randomized. The attrition rate was considered, so intention-to-treat (ITT) analysis was adopted to analyze data of the older adults who left the trial or did not complete the assessment procedures [50]. Being aware of its disadvantages, the last observation carried forward imputation as ITT method was performed [51].

For BPG variables when baseline and post-intervention assessment occurred some older adults were not able to finish assessment due to their foot physical conditions so for BPG consider for CG: *N* = 18 and EG: *N* = 20.

### 2.7. Ethical Considerations

All procedures performed in studies involving human participants were in accordance with the ethical standards of the institutional and/or national research committee and with the 1964 Helsinki declaration and its later amendments or comparable ethical standards. The Ethics Committee of the Health Sciences Research Unit: Nursing approved this Study (P455-09/2017). The formal authorization of each participating end-user organization was also obtained.

All the participants gave their written informed consent prior to participation in the study. After asking their consent, each participant choose one image from a set of images related to animals. This image worked as an identifier code in order to guarantee subject’s data confidentiality and identity. The coded data were only available to the research team.

CG participants were included in a waiting list to be engaged in a similar program post-trial under the responsibility of each partner institution and using the materials produced by the research team. All instruments were used with permission of the instrument ownership companies or authors.

## 3. Results

### 3.1. Baseline Assessment

As displayed in Table 1, the 44 older adults that participated in the study had a mean age of 80.54 ± 8.47 years (ranged from 65 to 96) and 29 (65.9%) of them were female. The most common civil state was a widow. In terms of education, the results revealed that 63.6% frequented school between 1–4 years and 18.2% never went to school, which demonstrates a low level of education among participants.

At baseline, the EG and CG were equivalent regarding sex, age, registration time in the end-user organization and the outcomes of interest (Table 1 and Table 2), including the outcome related to gait parameters. The baseline data is similar with ITT analysis or excluding the dropouts, therefore, only the ITT analysis results are presented here.

### 3.2. Post-Intervention Assessment

#### 3.2.1. Between-Group Analysis of Primary and Secondary Outcomes

Between-group analyses (Table 2) performed after intervention revealed statistically significant differences in depressive symptomatology measured through the GDS-10 (*U* = 147.00, *p* = 0.024), with the EG presenting a lower mean number of symptoms than the CG. Regarding effect size, it was revealed to be small (*effect size* = 0.118).

No other significant differences in primary and secondary outcomes were observed (for more detailed data see Table 2).

#### 3.2.2. Within-Group Analysis of Primary and Secondary Outcomes

Within-group analyses (for more details see Table 2 and Table 3) comparing differences in the study outcomes from baseline to post-intervention revealed that the EG significantly reduced depressive symptomatology assessed through the GDS-10 (*Z* = −2.27, *p* = 0.023), and risk of fall measured based on the gait and balance subtests from the Tinetti Test (Z = −2.61, *p* = 0.009), and the gait speed test (Z = −2.09, *p* = 0.037). For CG, within-groups analyses revealed that the activities of daily living worsen significantly from baseline to post-intervention (Z = −2.46, *p* = 0.021). No other significant differences were observed.

Noteworthy that are improvements ranging from 47.7% to 52.4% in cognition, depressive symptoms, and risk of fall, as well as between 55.6% and 66.7% in BPG. Percentage range in other variables is lower.

The effect size was calculated for variables with significant results. In the EG the effect size revealed to be small since the obtained values were between 0.2 ≤ *r* < 0.5. In the CG, the registered effect size was also small (Table 4).

A small attrition rate could be considered and can be justified by the before implementation’ procedures related to preparation of the setting, team, assessment, CI and participants recruitment may have influence the participants compliance.

## 4. Discussion

These study findings confirm the effectiveness of CI to improve psychological and physical capacities of frail older adults. The multimodal approach for the delay of geriatric comorbidities is well stated on the overall literature [52,53]. Some of the authors have even recommended a multimodal approach towards healthy and frail older adults, suggesting the implementation of combined interventions that already showed to have positive effects when singly implemented [12,23]. Besides recommendations, few studies focused on assessing the effectiveness of interventions that combine, simultaneously, cognitive stimulation and physical exercise programs, especially towards the frailty’ condition.

By purpose, the sample of this study included frail older adults from two different end-user organizations, it means day-centers and nursing homes. This choice, which induced the multicenter cluster randomized controlled trial approach, was made in order to get two subsets of frail older adults population, aiming to obtain a representative sample of the actual Portuguese geriatric panorama. The research team, supported by their expertise and knowledge, decided to combine these two contexts since they considered that the prevalence of the frail older adults in these contexts is quite high. Despite this fact, and considering the frailty concept introduced by the integral conceptual model of frailty proposed by Gobbens [6], the potential that the CI demonstrates is real, once some key results are revealed in promoting psychological and physical capacities of frail older adults.

After a 12 weeks CI implementation, there is an evident deterioration of cognitive performance in CG, when compared with the EG. These results are in concordance with the results found at the study conducted by [47] that singly implemented the original cognitive stimulation program among older adults from nursing homes. The gains in cognitive performance after receiving cognitive stimulation are also well stated on the literature [54,55]. According to a recent meta-analysis of the randomized controlled trails on the efficacy of cognitive stimulation in adults with dementia [56], improvements in cognitive functioning resulting from the intervention tend to be generalized, with either moderate or small effect. However, the neurological mechanism responsible for preserving the cognitive reserve remains unclear. On the other hand, the condition of cognitive frailty that constitutes an important risk factor for dementia [57,58] is suggested to be an ideal target for a secondary prevention of cognitive and functional impairment [2], since its effective monitoring allows the prevention of late-life cognitive disorders. Yet, further research is needed on the mechanisms underlying cognitive frailty and the relationships between this clinical condition and dementia to enable design of more suitable and more effective interventions.

Cognitive stimulation programs may also be an important non-pharmacological alternative for attenuating or delaying progression of neuropsychiatric symptoms, such as depression or anxiety, in older adults with dementia [59]. Curiously, there are some studies that delivered the similar cognitive stimulation program [47] as presented in this article and did not found statistical evidence on the reduction of the participants’ depressive symptomatology. This divergence on the obtained results could be explained by the strong social interaction that CI provides to older adults and also by the important biological effects that are associated to physical exercise combination [60]. The statistical evidence in terms of cognitive performance and mood (depressive symptomatology) lead us to affirm that in accordance with the Gobben’s Model of Frailty [6], CI is effective on the improvement of psychological capacities of frail older adults. The beneficial effects of the intervention’s group context in reducing frailty levels are also described by other authors [8]. According to these authors, the mechanisms that may be responsible for increasing the impact of intervention include commitment to co-participants, enjoyment and social interaction.

Considering now the physical outcomes, in comparison with the CG, the EG diminished the risk of fall based on gait and balance subtests of TI, and improved their gait speed. These results are also stated by the literature and may be associated to the fact that the multimodal approach induces several sensory inputs that contribute to the management of older adults proprioceptive notion, by stimulating visual and vestibular inputs [53]. The gains on overall balance and gait speed could also be associated to the gains on muscular mass, and the development of new blood vessels or nervous tissue, caused by the different types of exercise ministered.

The equipment and methodology used for gait analysis has progressed substantially in recent years. The information derived from plantar pressure and ground reaction forces measures is important in gait and posture research for diagnosing lower limb problems and injury prevention [61,62,63,64,65]. The assessment of plantar pressure and ground reaction forces measures also allow the analysis of the effects of a treatment [66,67,68]. As an example, patients with diabetic peripheral neuropathy have elevated plantar pressures and require more time in the stance-phase during gait [69]. The potential use of gait analysis in patients with diabetes has led to improvements in health care including treatment and prevention of ulceration and development of targeted exercise interventions. However, our results are nonetheless important regarding the main outcomes considering that higher plantar pressure values have been associated with a change in walking strategy and an increase in fall risk. Gait speed less than 0.8m/s is considered a predictor of poor health status, and impaired sub-clinical neurological and muscular [41,70,71]. For frail populations gait impairment is a common symptom of frailty and the literature defines a slow gait speed of ≤0.8 m/s (taking longer than 5 s to walk 4 m) as a cut-point for identify frailty among older adults [72].

Positive effects were also found in activities of daily living. The results on the BI are relevant to this study since they reveal that CI is sensitive to activities of daily living. Namely, our study has demonstrated that EG participants maintained their functional performance at post-test. Comparatively to EG, CG worsened their functional status. These results indicate that CI has potential to induce a transference effect between the physical and the psychological domains due to its complex characteristics of multifactorial approach [23,53]. Overall, this could mean that CI has a compensatory potential that emerge from the combination of both components, this is a structured and adapted cognitive stimulation program and a physical exercise program. This CI of 12 weeks does not allow to establish the potential effects of an exercise program in a group of frail older adults.

Considering the targeted population, the implementation of the developed multimodal CI seems to have positive effects in psychological or physical components of frailty, in accordance with the frailty model that was used to sustain this trial and the respective outcomes assessment. There is a variety of risk factors that are intimately related to the development of frailty among older adults. Curiously, the positive results found in this study coincide with the risk factors found among frail community-dwelling older adults, such as dependence on activities of daily living for physical factors and depressive symptomatology for psychological factors [73].

Further research should be focus on increasing the number of participants and age intervals. allowing the training effects of CI, between active and sedentary older adults, and with differentiated functional capacity. Moreover, longitudinal studies are required to test which gait parameters, either kinetics or kinematics are more sensitive to tailored exercise interventions.

We speculate that the long-term effectiveness of the CI will be established at the level of general mobility by means of adopting an active lifestyle, which can be translated into improved autonomy, independent living, and prevention of falls in beneficiaries of these interventions.

There are no references regarding frail older populations. To our knowledge, this is the first study investigating the effect of exercise on gait and exploring the assessment of gait parameters with frail older adults, namely plantar pressure and peak pressure.

## 5. Forces and Limitations

Complete blinding in this trial was problematic due to its design and the need for informed consent. CI implementers were aware of participant’s allocation to EG or CG assignment. Participants were also aware of if they were in the EG receiving the CI or if they are in the CG receiving standard care. However, the recruiters, outcome assessors, and those who analyzed data were blinded to the participant’s allocation group. The study has a relatively reduced sample size, which make conclusions regarding the effectiveness of CI less generalizable. However, due to the amount of variables that are involved, this trial involves a complexity that could induce more research questions for future studies. Future studies should have a larger sample size and different randomized procedure to investigate the long-term effects of CI in all considered outcomes, namely what it concerns to BPG. It also should measure the effect including a social CG, an only exercise CG, and an only cognitive stimulation CG, comparison with the CI, to truly understand intervention benefits.

## 6. Conclusions

The present trial validate the effectiveness of CI composed by adapted and structured cognitive and physical exercise programs to promote psychological and physical capacities of a sample of Portuguese frail older adults from day centers and nursing homes. The results based on the calculated effect size suggest that CI is effective mainly on the reduction of depressive symptomatology, and risk of fall and in the maintenance of cognitive performance. The findings of our study are relevant for the clinical practice as they suggest the efficacy of this type of non-pharmacological intervention that could be implemented as soon as possible on the opportunity window of the condition of frailty by professionals from different areas and by formal or informal caregivers. Future research should focus on the previously presented limitations and also expect the cost-effectiveness of this type of intervention, due to its affordable conditions.

## Figures and Tables

**Figure 1 ijerph-16-03125-f001:**
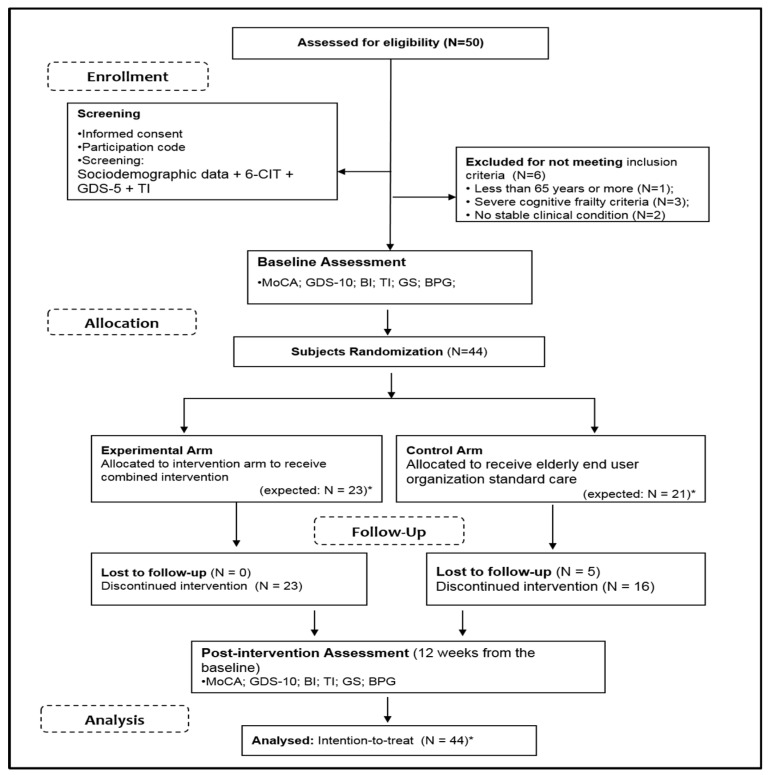
Consolidated standards of reporting clinical Trial (CONSORT) flow chart related to data collection. Note: 6-CIT—Six Item Cognitive Test; GDS-5—Geriatric Depression Scale with 5 items; MoCA—Montreal Cognitive Assessment; GDS-10—Geriatric Depression Scale with 10 items; BI–Barthel Index; TI–Tinetti Index; GS–Gait Speed; BPG–biomechanical parameters of gait. * For BPG variables CG: *N* = 18; EG: *N* = 20.

**Figure 2 ijerph-16-03125-f002:**
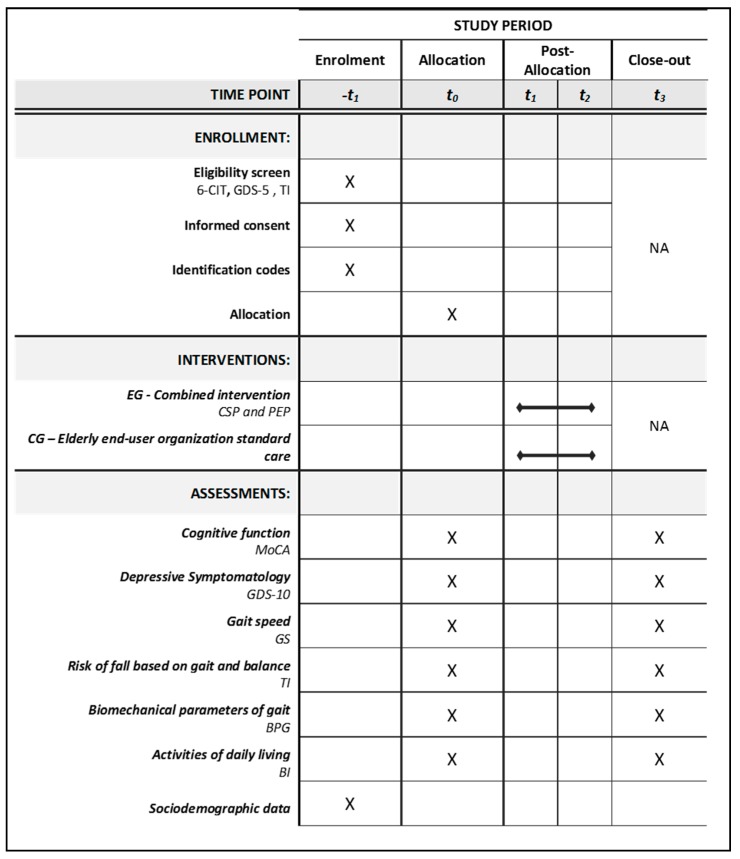
Schedule of enrollment, interventions and assessments based on SPIRIT 2013 Statement. Note: -t1—enrollment week; t0—allocation and baseline assessment; t1—Week 1; t2—Week 12; t3—Post-intervention assessment; 6-CIT—Six item cognitive Test; GDS-5—Geriatric depression scale with 5 items; EG—Experimental group; CSP—Cognitive stimulation program; AAA—Animal assisted activity; PEP—Physical exercise program; CG—Control group; MoCA—Montreal cognitive assessment; GDS-10—Geriatric depression scale with 10 items; GS—Gait speed; TI—Tinetti index; BPG—biomechanical parameters of gait; BI—Barthel index; NA—Not applicable.

**Figure 3 ijerph-16-03125-f003:**
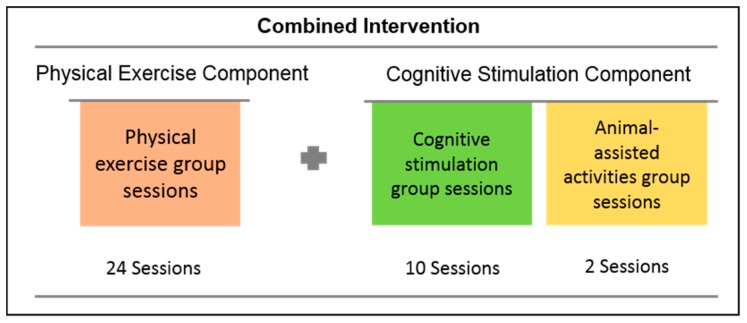
Combined intervention components.

**Table 1 ijerph-16-03125-t001:** Baseline Characteristics of the study sample.

Variables	Total Sample (*N* = 44)	Control Group (*n* = 21)	Experimental Group (*n* = 23)		
	***N*** **(%)**	***n*** **(%)**	***n*** **(%)**	**X*^2^***	***p***
**Sex**					
Male	15 (34,1)	5 (23.8)	10 (43.5)	1.89	0.213
Female	29 (65.9)	16 (76.2)	13 (56.5)	-	
	**M** **±** **SD (Median)**	**M** **±** **SD (Median)**	**M** **±** **SD (Median)**	***U***	***p***
**Age**	80.54 ± 8.47 (81)	82.04 ± 9.29 (85)	79.17 ± 7.59 (80)	192.500	0.249
**Registration time in the elderly end-user organization (months)**	49.97 ± 41.86 (36)	57.85 ± 46.66 (48)	42.78 ± 36.51 (24)	207.500	0.420

M—mean value; SD—Standard deviation.

**Table 2 ijerph-16-03125-t002:** Between group analysis—Mann Whitney Test.

Variables	Baseline					Post-Intervention				
	CG (*N* = 21)		EG (*N* = 23)		U (*p*)	CG (*N* = 21)		EG (*N* = 23)		U (*p*)
	M (SD)	MR	M (SD)	MR		M (SD)	MR	M (SD)	MR	
Cognitive performance (MoCA)	15.33 (4.83)	22.33	15.87 (4.88)	22.65	238.00 (0.934)	16.42 (6.49)	22.40	16.91 (4.98)	22.59	239.50 (0.962)
Depressive Symptomatology (GDS-10)	3.81 (2.84)	25.19	2.74 (2.70)	20.04	185.00 (0.180)	3 (2.24)	27.00	1.65 (2.06)	18.39	147.00 (0.024)
Risk of fall based on gait and balance (TI)	19.67 (6.38)	21.33	20.39 (7.28)	23.57	217.00 (0.564)	20.62 (6.61)	20.00	22.17 (6.69)	24.78	189.00 (0.211)
Gait Speed	0.59 (0.29)	22.68	0.55 (0.31)	21.41	216.50 (0.742)	2.46 (7.48)	22.23	0.64 (0.37)	21.80	225.50 (0.913)
Activities of daily living (BI)	0.82 (0.14)	20.26	0.84 (0.18)	24.54	194.50 (0.264)	0.75 (0.19)	18.83	0.82 (0.19)	25.85	164.50 (0.069)
Maximum force all foot right *	100.87 (7.16)	20.95	104.02 (19.79)	20.05	191.00 (0.808)	101.69 (6.73)	18.28	112.99 (44.15)	21.48	158.00 (0.382)
Maximum force all foot left *	90.52 (29.96)	18.60	105.26 (20.87)	22.40	162.00 0.304)	102.65 (6.76)	18.94	151.66 (217.56)	20.90	170.00 (0.592)
Peak pressure all foot right *	445.53 (169.04)	21.05	481.66 (302.16)	19.95	189.00 (0.766)	452.32 (193.34)	19.28	524.31 (303.94)	29.62	176.00 (0.714)
Peak pressure all foot left *	410.46 (175.93)	19.08	520.51 (278.35)	21.93	171.50 (0.441 *)	477.40 (198.84)	19.50	545.35 (290.31)	20.43	180.00 (0.800)

Note. M—Mean value; SD—Standard deviation; MR—Mean rank; * For BPG variables when baseline and post-intervention assessment occurred some older adults were not able to finish assessment due to their foot physical conditions so for BPG consider for CG: N = 18 and EG: N = 20.

**Table 3 ijerph-16-03125-t003:** Within groups comparison with ranks for Wilcoxon test.

Variables	CG (N = 21)						EG (*N* = 23)					
	Status **	N	%	Mean Rank	*Z*	*p*	Status **	*N*	%	Mean Rank	*Z*	*p*
Cognitive Performance (MoCA)	a	10	47.6	8.50	−1.43	0.153	a	14	60.9	9.00	−1.78	0.075
	b	5	23.8	7.00			b	4	17.4	11.25		
	c	6	28.6				c	5	21.7			
Depressive Symptomatology (GDS-10)	a	10	47.7	7.65	−1.52	0.130	a	11	47.8	9.05	−2.27	0.023
	b	4	19.0	7.13			b	4	17.4	5.13		
	c	7	33.3				c	8	34.8			
Risk of Fall Based on Gait and Balance (Tinetti)	a	11	52.4	8.50	−1.33	0.184	a	18	78.26	10.56	−2.61	0.009
	b	5	23.8	8.50			b	3	13.04	13.67		
	c	5	23.8				c	2	8.7			
Gait Speed	a	6	30	11.00	−0.50	0.618	a	15	65.2	11.70	−2.09	0.037
	b	11	55	7.91			b	6	26.1	9.25		
	c	3	15				c	2	8.7			
Activities of Daily Living (Barthel Index)	a	2	9.5	6.25	−2.31	0.021	a	3	13.0	7.17	−1.03	0.303
	b	11	52.4	7.14			b	8	34.8	5.56		
	c	8	38.1				c	12	52.2			
Maximum Force all Foot Left *	a	12	66.7	9.58	−2.43	0.015	a	14	70	10.79	−2.254	0.024
	b	4	22.2	5.25			b	5	25	7.80		
	c	2	11.1				c	1	5			
Maximum Force all Foot Right *	a	10	55.6	8.90	−1.086	0.278	a	16	80	9.31	−2.173	0.030
	b	6	33.3	7.83			b	3	15	13.67		
	c	2	11.1				c	1	5			
Peak Pressure All Foot Left *	a	11	61.1	8.73	−2.045	0.041	a	11	55	11.27	−1.167	0.243
	b	4	22,2	6.00			b	8	40	8.25		
	c	3	16.7				c	1	5			
Peak Pressure All Foot Right *	a	11	61.1	8.09	−1.086	0.278	a	13	65	10.12	−2.004	0.045
	b	5	27.8	9.40			b	5	25	7.90		
	c	2	11.1				c	2	10			

Note. * For BPG variables’ when baseline and post-intervention assessment occurred some older adults were not able to finish assessment due to their foot physical conditions so for BPG consider for CG: N = 18 and EG: N = 20. ** a = improved; b = deteriorated; c = maintained.

**Table 4 ijerph-16-03125-t004:** Effect size values.

Varaibles	Control Group		Experimental Group	
	*Z*	*Effect Size* (*r*)	*Z*	*Effect Size* (*r*)
Depressive Symptomatology (GDS-10)			−2.27	0.33469
Risk of Fall Based on Gait and balance (Tinetti)			−2.61	0.38482
Gait Speed			−2.09	0.30815
Activities of Daily Living (Barthel Index)	−2.31	0.35644		
Maximum Force All Foot Left *	−2.43	0.40500	−2.254	0.35639
Maximum Force All Foot Right *			−2.173	0.34358
Peak Pressure All Foot Left *	−2.045	0.33400		
Peak Pressure All Foot Right *			−2.004	0.31686

Note. * For BPG variables when baseline and post-intervention assessment occurred some older adults were not able to finish assessment due to their foot physical conditions so for BPG consider for CG: N = 18 and EG: N = 20.

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
