# Peer review of "Effectiveness of a Combined Intervention on Psychological and Physical Capacities of Frail Older Adults: A Cluster Randomized Controlled Trial"

_ijerph, 2019, doi:10.3390/ijerph16173125_

Round 1
Reviewer 1 Report
This is a well done and written scientific work.
I have two suggestions:
replacing the term "humor" with the more appropriate and used "mood" throughout the text; adding "Animal-Assisted Activities" to key words to facilitate those readers interested in this kind of intervention in finding this work. Repeated, generic or not relevant key words like "aged" "cognition" and "independent living" should be removed. "Exercise" may be replaced with "physical exercise".Author Response
Reviewer:
I have two suggestions:
replacing the term "humor" with the more appropriate and used "mood" throughout the text; adding "Animal-Assisted Activities" to key words to facilitate those readers interested in this kind of intervention in finding this work. Repeated, generic or not relevant key words like "aged" "cognition" and "independent living" should be removed. "Exercise" may be replaced with "physical exercise".
Author response: Suggestions were accepted and incorporated in the manuscript.
Please see the attachment.

Reviewer 2 Report
Effectiveness of a combined intervention on psychological and physical capacities of frail older adults: a cluster randomized controlled trial
Manuscript number IJERPH (ISSN 1660-4601)
General: Well written and organized manuscript.Title Edit the use of frail from describing the older adults, since frailty was not seen as an eligibility requirement. Consider instead frailty as a description of the outcomes: “psychological and physical frailty capacities”. Introduction: The discussion about frailty definition and assessment can be condensed. Since physical exercise has been shown to improve both physical and cognitive/emotional health status, it is necessary to highlight here why there is a need to offer a combined intervention. Methods: Recruitment of end-user organizations and participants needs to be discussed more clearly. Some details about how the assessment battery was facilitated is needed. Unclear how and if frailty status was verified into eligibility criteria. Eligibility for end-user organization: “not offer as standard care adapted and structured cognitive stimulation programs or physical exercise programs”, seems impractical and inhumane on part of the organization. Also, the stipulation as part of participation: “To decrease contamination, CG participants were asked not to carry out activities similar to intervention program,” also is inappropriate for humanity reasons. Unclear why frailty is not an eligibility criterion if frail older adults is the central group. Results No attrition, great! Discuss how. Need better understanding of the Status provided in Table 3: improved, deteriorated, and maintained, specifically how were they assessed and analyzed. Also, percent values for status in Table 3 would be more helpful. Discussion Need to discuss that end-user organizations need to make CI type interventions part of their standard of care, not as a treatment/intervention specialized service. And note that those that do not offer any cognitive stimulation and physical exercise programs to all their clients, including frail clients, are providing a disservice. In the limitation, need to discuss the need for a social control group, exercise only control group, and cognitive stimulation only control group, comparison with the CI intervention, to truly understand intervention benefits.
